

# Respiratory syncytial virus genotypes NA1, ON1, and BA9 are prevalent in Thailand, 2012–2015

Ilada Thongpan, John Mauleekoonphairoj, Preeyaporn Vichiwattana, Sumeth Korkong, Rujipat Wasitthankasem, Sompong Vongpunsawad and Yong Poovorawan

Center of Excellence in Clinical Virology, Faculty of Medicine, Chulalongkorn University, Bangkok, Thailand

## ABSTRACT

Respiratory syncytial virus (RSV) causes acute lower respiratory tract infection in infants and young children worldwide. To investigate the RSV burden in Thailand over four consecutive years (January 2012 to December 2015), we screened 3,306 samples obtained from children ≤5 years old with acute respiratory tract infection using semi-nested reverse-transcription polymerase chain reaction (RT-PCR). In all, 8.4% (277/3,306) of the specimens tested positive for RSV, most of which appeared in the rainy months of July to November. We then genotyped RSV by sequencing the G glycoprotein gene and performed phylogenetic analysis to determine the RSV antigenic subgroup. The majority (57.4%, 159/277) of the RSV belonged to subgroup A (RSV-A), of which NA1 genotype was the most common in 2012 while ON1 genotype became prevalent the following year. Among samples tested positive for RSV-B subgroup B (RSV-B) (42.6%, 118/277), most were genotype BA9 (92.6%, 87/94) with some BA10 and BA-C. Predicted amino acid sequence from the partial G region showed highly conserved N-linked glycosylation site at residue N237 among all RSV-A ON1 strains (68/68), and at residues N296 (86/87) and N310 (87/87) among RSV-B BA9 strains. Positive selection of key residues combined with notable sequence variations on the G gene contributed to the continued circulation of this rapidly evolving virus.

Corresponding author
Yong Poovorawan,
Yong.P@chula.ac.th

## INTRODUCTION

Human respiratory syncytial virus (RSV) is a major cause of severe respiratory infections in infants and young children (*Nair et al., 2010*). Although RSV infections in healthy adults are generally limited to the upper respiratory tract, symptoms in infants and children include wheezing, cough, bronchiolitis and pneumonia (*Hall, Simoes & Anderson, 2013*). RSV is a member of the newly assigned genus *Orthopneumovirus* in the family *Pneumoviridae* (*Afonso et al., 2016*). It has a single-stranded negative-sense RNA genome of approximately 15.2 kb, which encodes 11 viral proteins (*Collins, Fearns & Graham, 2013*). The viral envelope glycoproteins G and F are important for the receptor attachment and viral fusion with the target cell, respectively. Both surface glycoproteins can induce the host immune response (*Wagner et al., 1989*).

RSV can be divided into two distinct subgroups (RSV-A and RSV-B) (*Mufson et al., 1985*). Subgroups can be further divided into genotypes based on the gene sequence variability of the second hypervariable region (HVR2) located on the distal third region of the G gene (*Melero, Mas & McLellan, 2017*). As a result, there are 13 RSV-A genotypes: GA1 to GA7 (*Peret et al., 2000*), SAA1 (*Venter et al., 2001*), NA1 to NA4 (*Shobugawa et al., 2009*; *Cui et al., 2013*), and ON1 (*Eshaghi et al., 2012*). To date, 22 RSV-B genotypes are GB1 to GB4 (*Peret et al., 2000*), SAB1 to SAB4 (*Venter et al., 2001*; *Arnott et al., 2011*), URU1 and URU2 (*Blanc et al., 2005*), BA1 to BA10 (*Trento et al., 2006*), BA-C (*Cui et al., 2013*), and THB (*Auksornkitti et al., 2014*).

Within the past decade, a new genotype of RSV-A identified in Ontario, Canada (ON1) has emerged (*Eshaghi et al., 2012*). This genotype is characterized by a 72-nucleotide duplication in the HVR2. ON1 has since been detected in Europe, Africa, Asia, and the Americas (*Cui et al., 2013*; *Valley-Omar et al., 2013*; *Pierangeli et al., 2014*). Meanwhile, an RSV-B genotype identified in Buenos Aires, Argentina (BA) similarly contained a 60-nucleotide duplication in the HVR2 (*Trento et al., 2003*). BA has subsequently became so prevalent globally that it largely replaced other RSV-B genotypes (*Trento et al., 2010*). Gene insertion seen in both RSV-A and RSV-B HVR2 may have affected the antigenicity of the G protein and enabled viral escape from the host immunity, thus mediating their predominance in recent years.

Due to the lack of antivirals or approved vaccines for RSV (*Graham & Anderson, 2013*), epidemiological surveillance studies are crucial in understanding RSV transmission and spread. Our previous study described the molecular epidemiology of RSV strains in circulation in Thailand between 2010 and 2011 (*Auksornkitti et al., 2014*). Here, we characterize recent incidence of RSV infection in Thailand from 2012 to 2015 and examined how the G gene of the circulating strains of RSV are changing, which may explain the persistence of RSV circulation in this region.

## MATERIALS AND METHODS

### Samples

This study was approved by the Institutional Review Board of the Faculty of Medicine, Chulalongkorn University (IRB 609/59). The need for consent was waived by the IRB since this study involved the use of anonymous residual specimens. In all, 3,306 archived and convenient specimens from January 2012 to December 2015 previously subjected to routine respiratory viral screening in our laboratory were tested for the presence of RSV. These samples comprised nasopharyngeal aspirates, nasal swabs, and throat swabs from individuals ≤5 years of age (mean age 2.4 years, male-to-female ratio of 1 to 0.8). Patients presented influenza-like illness (defined as the presence of body temperature >38 °C and respiratory tract symptoms such as runny nose, cough, sore throat, and difficulty breathing) or were diagnosed with probable RSV infection by the physicians. Samples were from Bangkok ($n = 1,222$ from King Chulalongkorn Memorial Hospital and Bangpakok 9 International Hospital) and from Khon Kaen province ($n = 2,084$ from Chum Phae Hospital).

## RNA extraction, reverse transcription, and PCR amplification

RNA was extracted using the Viral Nucleic Acid Extraction Kit (RBC Bio-science) and cDNA synthesized using an ImProm-II Reverse Transcription System (Promega) according to the manufacturer's instructions. Briefly, RNA and random hexamer primers were incubated at 70 °C for 5 min, followed by extension for 2 h at 42 °C and inactivation at 70 °C for 15 min. The amplification of the partial G (inclusive of the HVR2) and F genes was performed using semi-nested PCR as previously described (*Auksornkitti et al., 2014*). First-round amplification involved initial denaturation at 94 °C for 3 min; 40 cycles of denaturation at 94 °C for 20 s, annealing at 55 °C for 20 s, and elongation at 72 °C for 90 s; and a final extension at 72 °C for 10 min. Identical amplification parameters were carried out in the second-round PCR for 30 cycles. The PCR amplicons (approximately 840 bp for RSV-A and 720 bp for RSV-B) were visualized by 2% agarose gel electrophoresis and purified PCR products were sequenced.

## Sequence comparison and phylogenetic analysis

Nucleotide sequences were visualized using Chromas Lite (v2.01), assembled using SeqMan (DNASTAR), analyzed using the Basic Local Alignment Search Tool (BLAST) program (http://blast.ncbi.nlm.nih.gov/Blast.cgi), and aligned using BioEdit (v7.0.9.0) and ClustalW. For genotype assignment, 211 strains of RSV-A ($n = 117$) and RSV-B ($n = 94$) in which the partial nucleotide sequences for the G gene were successfully obtained were subjected to phylogenetic analysis. RSV-A strain A2 (genotype GA1, GenBank accession number M74568) and RSV-B strain B1 (genotype GB1, GenBank accession number AF013254) served as reference strains. Phylogenetic trees were constructed using the neighbor-joining method implemented in MEGA (v6.0) with 1,000 bootstrap replicates (*Tamura et al., 2013*). Differences between (inter-) and within (intra-) genotypes were evaluated by pairwise nucleotide distance (*p*-distance) calculations. All RSV sequences obtained from this study were deposited in the GenBank database under the accession numbers KY327937–KY328054 (RSV-A) and KY328055–KY328148 (RSV-B).

## Predictions of glycosylation sites and positive selection analysis

Deduced amino acid sequences for the partial G gene HVR2 were analyzed for potential N-linked glycosylation sites using the online tool NetNGlyc version 4.0 (http://www.cbs.dtu.dk/services/NetNGlyc/). Tests for positive selection were conducted using single-likelihood ancestor counting (SLAC), fixed-effects likelihood (FEL), internal fixed-effects likelihood (IFEL), and random-effects likelihood (REL) models on the Datamonkey server (http://www.datamonkey.org/), with the ratio of divergence at nonsynonymous and synonymous sites (dN/dS) calculated using the SLAC model. To avoid an excessive false-positive rate, sites with SLAC, FEL and IFEL *p*-values of <0.1 were accepted as under positive selection. Sites with REL Bayesian factors of >50 were also accepted as positively selected sites.

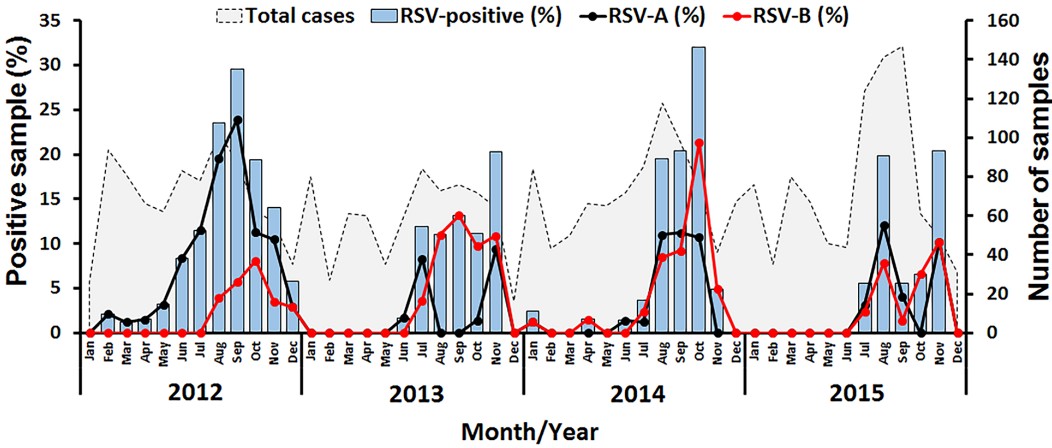

**Figure 1** **Seasonal distribution of RSV detected between 2012 and 2015 in this study.** The number of samples examined by month is shown as gray peaks (right scale). RSV positive rates (bar graphs) for RSV-A (black line) and RSV-B (red line) are also shown (left scale).

**Table 1** **RSV identified between 2012 and 2015 from patients with acute respiratory tract infections.**

| Year | No. of samples | RSV-positive (%) | Groups | |
|------|----------------|------------------|--------|--------|
| | | | RSV-A (%) | RSV-B (%) |
| 2012 | 834 | 94 (11.3) | 77 (81.9) | 17 (18.1) |
| 2013 | 707 | 50 (7.1) | 15 (30) | 35 (70) |
| 2014 | 864 | 86 (8.8) | 35 (40.7) | 41 (59.3) |
| 2015 | 901 | 57 (6.3) | 32 (56.1) | 25 (43.9) |
| Total | 3,306 | 277 (8.4) | 159 (57.4) | 118 (42.6) |

## RESULTS

### RSV prevalence

In this study, RSV was identified in 8.4% (277/3,306) of all samples tested, of which 57.4% (159/277) were RSV-A and 42.6% (118/277) were RSV-B (Table 1). Cyclical pattern was reflected by the predominance of RSV-A in 2012, RSV-B in 2013 and 2014, and the subsequent return of RSV-A predominance in 2015. Co-infection with RSV-A and RSV-B was extremely rare (0.4%, 1/277). RSV infection appeared to peak in the rainy months of July to November with the highest incidence (32%, 24/75) in October 2014 (Fig. 1).

### Genotyping and phylogenetic analysis

Analysis of the partial G gene sequence showed that 41% (48/117) of RSV-A were genotype NA1 (Fig. 2). This genotype was predominant in 2012, but was relatively absent thereafter. An additional 58.1% (68/117) of RSV-A were ON1, which co-circulated with NA1 in 2012 and remained the only RSV-A genotype detected from 2013 onward. Only one sample was NA3, which appeared in 2012. For RSV-B, most were BA9 (92.6%, 87/94) (Fig. 3). The remainders were BA10 (4.3%, 4/94) and BA-C (3.2%, 3/94), which only appeared in

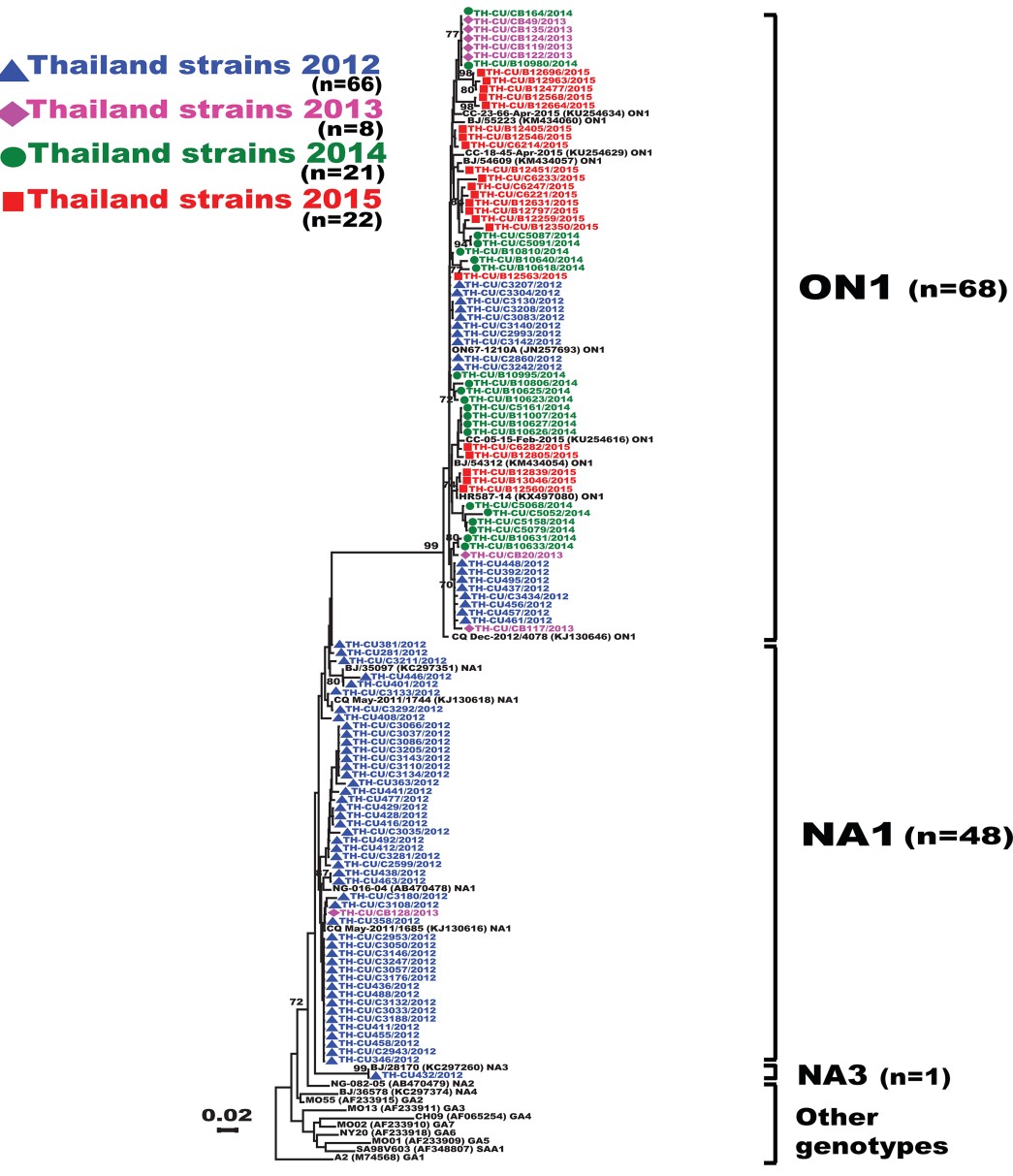

**Figure 2  Phylogenetic tree of RSV-A based on the partial nucleotide sequences of the second hypervariable region (HVR2) of the G gene.** The phylogenetic tree was constructed using the neighbor-joining algorithm in MEGA 6.0 software. Only bootstrap values >70% are displayed at the branch nodes. Years of virus isolation are color-coded for 2012 (blue triangle), 2013 (pink diamond), 2014 (green circle), and 2015 (red square). Other genotypes consist of GA1 to GA7, SAA1, NA2 and NA4. The scale bar represents the number of nucleotide substitutions per site between close relatives. The number of strains are in parentheses.

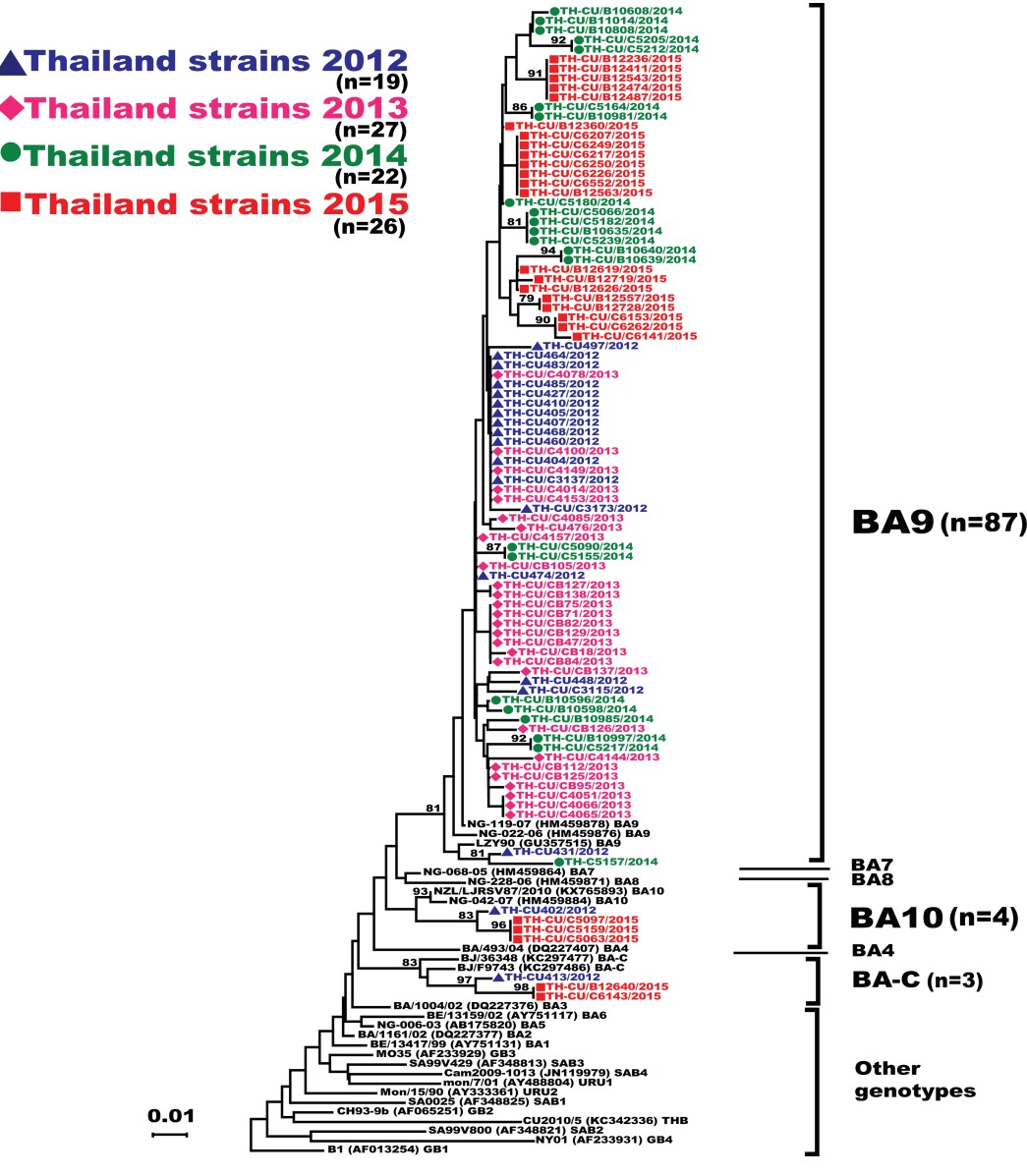

**Figure 3  Phylogenetic tree of the RSV-B nucleotide sequences based on the second hypervariable region (HVR2) of the G gene.** The phylogenetic tree was constructed using the neighbor-joining algorithm in MEGA 6.0 software. Bootstrap values >70% are displayed at the branch nodes. Years of virus isolation are color-coded for 2012 (blue triangle), 2013 (pink diamond), 2014 (green circle), and 2015 (red square). Other genotypes consist of GB1 to GB4, SAB1 to SAB4, URU1, URU2, THB, BA1 to BA3, BA5 and BA6. The scale bar represents the number of nucleotide substitutions per site between close relatives. The number of strains are in parentheses.

2012 and 2015. One sample (TH-CU448) was co-infected with RSV-A genotype ON1 and RSV-B genotype BA9.

For RSV-A, we found an equally high degree of intra-genotype diversity between NA1 and ON1 (Table S1). The *p*-distances within the NA1 and ON1 genotypes ranged from 0.003 to 0.06 (mean = 0.03). In contrast, the inter-genotype *p*-distance between the ON1

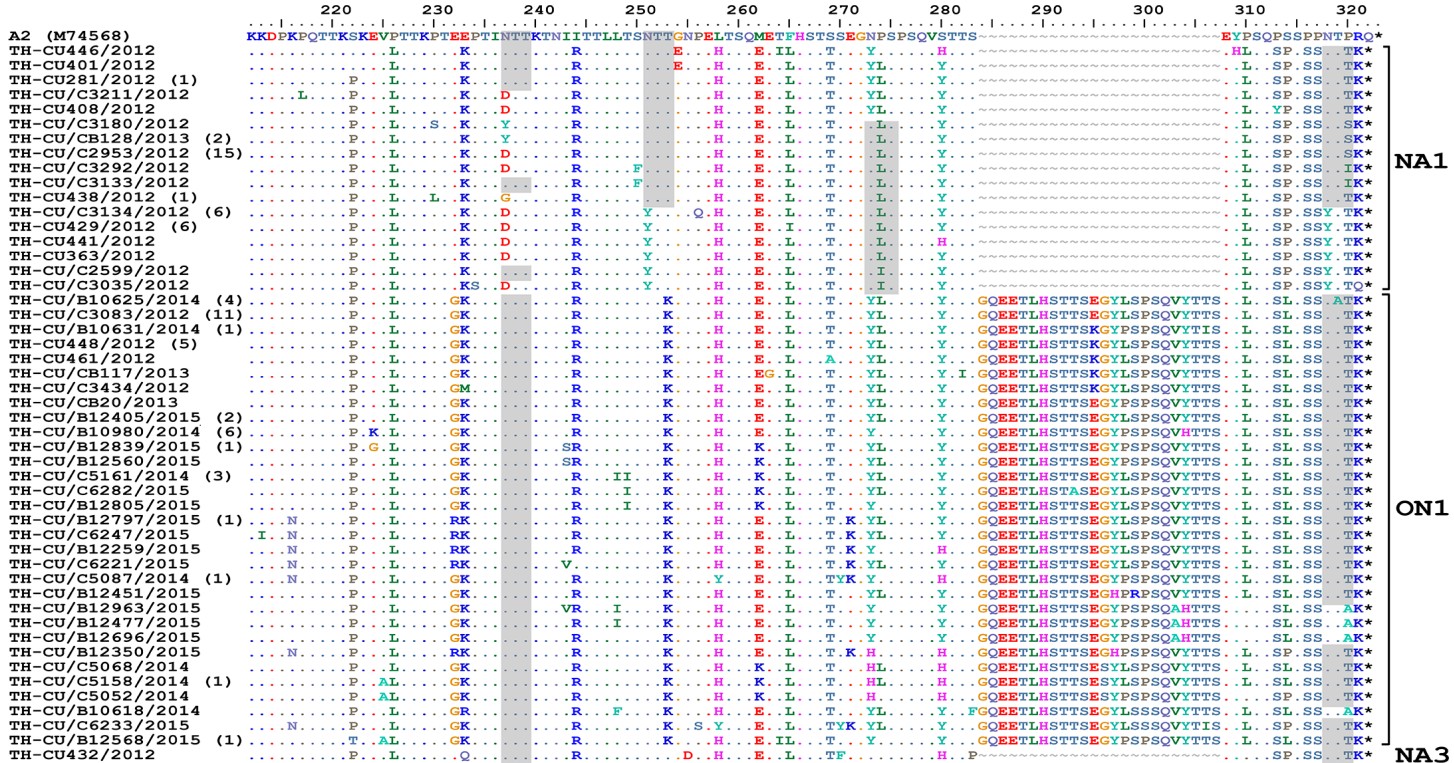

**Figure 4  Alignment of the deduced amino acid residues encoding the second hypervariable region (HVR2) of the G protein RSV-A strains identified in this study compared to the reference A2 strain.** The strain genotypes (NA1, ON1, and NA3) are indicated on the right (brackets). The number of additional strains with identical sequence as shown is indicated in parentheses to the right of the strain name. The presence of potential N-linked glycosylation sites are shaded. Asterisks indicate the positions of stop codons. Dots indicate identical residues. Tildes indicate missing residues.

and NA1 was lower (0.050), consistent with their close genetic relatedness. For RSV-B, the p-distance among the BA10 strains was 0.010. However, diversity among the BA9 strains was higher (p-distances ranged from 0 to 0.067). BA9 and BA10 were more closely related to one another (p-distance of 0.062) than to BA-C (p-distances of 0.090 and 0.080, respectively).

## Mutational analysis

The predicted amino acid sequences of RSV-A strains obtained in this study corresponded to residue position 212–298 (based on A2 reference strain numbering). As expected, amino acid alignment revealed that all ON1 possessed 24-residue insertion, of which 23 residues are duplication (Fig. 4). Potential N-linked glycosylation site appeared most conserved at N237 among all ON1 (100%, 68/68) and less so for N318 (94.1%, 64/68), but was absent at N251 and N273. Although most (87.5%, 42/48) NA1 strains lost the N237 site due to N237D/Y/G substitution, the majority (62.5%, 30/48) also gained an additional potential glycosylation site at N318. It is noteworthy that NA1 in general possessed either the glycosylation site at N237 or N273, but not both. The lone NA3 identified in this study possessed glycosylation sites at positions N237, N251, and N273.

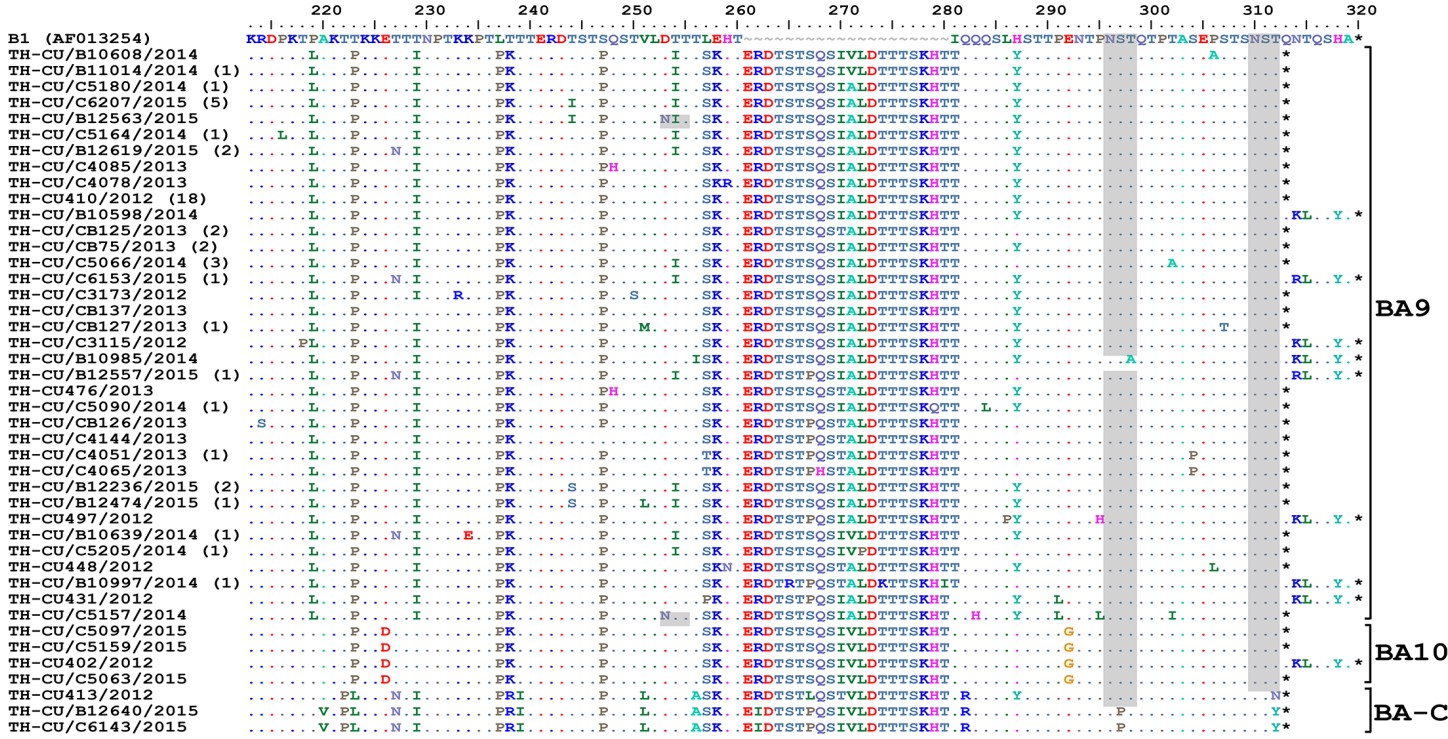

**Figure 5** **Alignment of the deduced amino acid residues encoding the second hypervariable region (HVR2) of the G protein RSV-B strains identified in this study compared to the reference B1 strain.** The strain genotypes (BA9, BA10, and BA-C) are indicated on the right (brackets). The number of additional strains with identical sequences as shown is indicated in parentheses to the right of the strain name. The presence of potential N-linked glycosylation sites are shaded. Asterisks indicate the positions of stop codons. Dots indicate identical residues. Tildes indicate missing residues.

Alignment of RSV-B amino acid sequence with the reference strain B1 revealed an insertion of 20 residues characteristic for the BA genotypes (Fig. 5). Two potential N-linked glycosylation sites, N296 and N310 (based on B1 numbering), were well-conserved in BA9 and BA10, but not BA-C due to T312Y/N substitution. A gain of glycosylation site at N253 was seen in some BA9 strains (TH_CU/B12563/2015 and TH_CU/C5157/2014).

## Selection pressure analysis

The high substitute rate typically seen in the genes encoding viral surface glycoproteins led us to analyze possible selective pressure on the G gene. Using the SLAC model, we found that the overall dN/dS ratios for NA1, ON1, and BA9 were 1.13, 1.11, and 0.65, respectively (Table 2). Four residues (233, 274, 298, and 314) of ON1 were under selection. Among these, the strongly site-specific P274L and L298P substitutions were identified in at least three selection models. Amino acid position 237 of NA1 and position 244 of BA9 were also likely under positive selection pressure, but these sites were identified only by single selection models.

**Table 2  Selection pressure analysis by single-likelihood ancestor counting (SLAC), fixed-effects likelihood (FEL), internal fixed-effects likelihood (IFEL), and random effects likelihood (REL) methods.**

| Genotype (Mean dN/dS) | Codon | SLAC | | FEL | | IFEL | | REL | |
|---|---|---|---|---|---|---|---|---|---|
| | | dN/dS | *p*-value | dN/dS | *p*-value | dN/dS | *p*-value | dN/dS | Bayesian factor |
| NA1 (1.13) | 237 | – | – | – | – | – | – | 6.665 | 97643.700 |
| | 273 | – | – | – | – | 66.265 | 0.084 | – | – |
| | 296 | – | – | – | – | 168.21 | 0.078 | – | – |
| ON1 (1.11) | 233 | – | – | 17.361 | 0.064 | | | 1.778 | 50.153 |
| | 274* | 10.685 | 0.050 | 27.016 | 0.075 | 36.324 | 0.067 | – | – |
| | 298* | 21.605 | 0.002 | 55.119 | 0.012 | 58.489 | 0.023 | 14.822 | 14894.500 |
| | 314 | 8.800 | 0.080 | – | – | | | – | – |
| BA9 (0.65) | 244 | – | – | – | – | 67.675 | 0.040 | – | – |

**Notes.**
*Indicate strong site-specific substitutions with at least three selection models.

## DISCUSSION

The muti-year study to investigate the prevalence and genetic diversity of RSV circulating in Thailand was undertaken because of the relative lack of recent prevalence data on RSV infection in this region. The overall frequency of RSV (8.4%) found in respiratory samples in this study was similar to the rates from previous studies in Thailand of 8.1% to 8.9% (*Fry et al., 2010*; *Naorat et al., 2013*). The prevalence of RSV in Thailand appears to be moderate and similar to those reported in Senegal (11.4%) (*Fall et al., 2016*), Colombia (8.9%) (*Barbosa Ramirez et al., 2014*), and the United States (7.7%) (*Fowlkes et al., 2014*). Although lower prevalence rates have been reported in Korea (2.7%) (*Noh et al., 2013*) and Brazil (2.4%) (*Bellei et al., 2008*), RSV infection is reportedly frequent in western China (23.7%) (*Hu et al., 2017*), the Philippines (19.3%–28.1%) (*Ohno et al., 2013*; *Malasao et al., 2015*), Vietnam (23%) (*Yoshida et al., 2010*), Indonesia (23%) (*Simões et al., 2011*), and Malaysia (18.4%) (*Chan et al., 2002*). A recent single hospital study in Laos identified RSV in as many as 41% of infected children (*Nguyen et al., 2017*). Unfortunately, objective comparison of the true regional prevalence of RSV is often hindered by the differences in the RSV detection rates largely due to sample size and the period of surveillance among these studies. Nevertheless, it is clear that RSV infection contributes to a significant portion of respiratory tract infection among young children.

In our study, RSV-A (57.4%) occurred more frequently than RSV-B and this observation is consistent with previous reports from other countries (*Ohno et al., 2013*; *Pretorius et al., 2013*; *Esposito et al., 2015*). Several studies including ours have reported alternating epidemics of RSV-A and RSV-B over time (*Cui et al., 2013*; *Ohno et al., 2013*; *Gilca et al., 2006*; *Zlateva et al., 2007*). This is not surprising since multiple genotypes of RSV-A and RSV-B are known to co-circulate in a single season and emerging genotypes may replace previously predominant ones in subsequent years (*Pretorius et al., 2013*). It is hypothesized that the periodic shifts in the predominant RSV subgroup are driven by the dynamics of population immunity and subgroup-specific herd immunity (*Botosso et al., 2009*), which

is analogous to periodic lineage shifts of the influenza B virus over time (*Tewawong et al., 2017*). Although co-infections by both RSV-A and RSV-B have previously been described in multiple regions (*Esposito et al., 2015*; *Parveen et al., 2006*; *Kouni et al., 2013*), it was relatively rare in our study.

Variability in RSV infection depends on geographical region, season, and year. In Southeast Asia, RSV incidence appears to peak during the rainy season (July to November), while relatively low rate of infection occurs during the hot and dry months (*Arnott et al., 2011*; *Nguyen et al., 2017*; *Yoshida et al., 2010*). This is in contrast to the peak incidence in countries with temperate climates whereby RSV infections are more frequent in the winter months (*Gilca et al., 2006*; *Panayiotou et al., 2014*; *Esposito et al., 2015*). For RSV-A, NA1 was the major genotype found in our previous study conducted between 2010 and 2011 (*Auksornkitti et al., 2014*) and was also seen here in 2012. More recently, ON1 has displaced NA1 to become the predominantly circulating RSV-A strain (*Eshaghi et al., 2012*; *Valley-Omar et al., 2013*; *Pierangeli et al., 2014*; *Kim et al., 2014*), although it does not appear to cause more severe disease than other genotypes (*Panayiotou et al., 2014*; *Viegas, Goya & Mistchenko, 2016*). Our study confirms findings of previous studies in that NA1 and ON1, but not NA3, are frequently identified in Southeast Asia (*Arnott et al., 2011*; *Yoshihara et al., 2016*).

Between 2012 and 2015, RSV-B genotypes BA9, BA10, and BA-C were frequently identified. BA9 appeared in Thailand beginning in 2010 and is now the predominant genotype, while other RSV-B genotypes appeared only sporadically. BA9 and BA10 are commonly identified in Southeast Asia (*Arnott et al., 2011*; *Tran et al., 2013*; *Tuan et al., 2015*). It is noteworthy that the BA genotypes, like the ON1, only recently emerged to replace all other RSV-B strains (*Trento et al., 2010*). Furthermore, the BA genotypes also possessed duplication in the HVR2, which is hypothesized to enhance RSV transmissibility by changing the antigenic epitope and facilitate immune evasion (*Viegas, Goya & Mistchenko, 2016*).

The relatively high genetic variability associated with the HVR2 was confirmed by the alignment of the predicted amino acid sequences of strains from both RSV subgroups. Of significance was the observed changes in the potential N-linked glycosylation sites, which can alter the antigenic characteristics of the viral surface protein by masking of specific epitopes from the host immunity (*Collins & Graham, 2008*; *Garcia-Beato et al., 1996*). Thus, mutations leading to novel glycosylation sites on viral surface glycoproteins may enable immune evasion. Glycosylation site at N237 was found to be highly conserved in all ON1 strains examined in this study. In contrast, only some NA1 strains possessed N237. A gain in glycosylation at N318 was also observed for some NA1 and ON1. One reason in which NA1 in our study demonstrated more heterogeneity at positions 251 and 318 than in other studies may be attributed to our multiple years of surveillance and larger sample size. The observed variations in the potential N-linked glycosylation sites tend to be genotype-specific, and coincidentally most of the positively selected sites in the G protein were in close proximity to and, in some cases, overlapped with, predicted glycosylation motifs (i.e., at amino acid positions 237 and 274). This suggests that positive selection based on N-linked glycosylation sites may play an important role in driving viral diversity (*Do et al., 2015*) as was shown with residue 274, which is located within an antigenic

site and is under positive selection (*Eshaghi et al., 2012*; *Cane, 1997*; *Ahmed et al., 2016*; *Malasao et al., 2015*). In summary, understanding the genetic variability in the RSV G gene is important in evaluating its evolving pathogenicity and transmission potential.

## CONCLUSION

As the G protein of RSV continues to accumulate novel sequence variations, which alter potential N-linked glycosylation patterns and exert positive selection of critical residues, results from this study provide additional knowledge regarding RSV infection patterns and viral diversity crucial for RSV vaccine development.

## ACKNOWLEDGEMENTS

We thank the administrative and laboratory staff of the Center of Excellence in Clinical Virology and King Chulalongkorn Memorial Hospital.

### Funding

This work was supported by the National Research Council of Thailand, the National Research University Project, Office of Higher Education Commission (NRU-59-002-HR), the Research Chair Grant from the National Science and Technology Development Agency (P-15-50004), the Outstanding Professor of Thailand Research Fund (DPG5480002), Chulalongkorn University Ratchadaphiseksomphot Endowment Fund (RES560530093) and Centenary Academic Development Project (CU56-HR01), the Center of Excellence in Clinical Virology (GCE 58-014-30-004), and King Chulalongkorn Memorial Hospital. This research was also supported by the Rachadapisek Sompote Fund of Chulalongkorn University to Rujipat Wasitthankasem, the 100th Anniversary Chulalongkorn University Fund for doctoral scholarship and the Overseas Research Experience Scholarship for Graduate Student to Ilada Thongpan. The funders had no role in study design, data collection and analysis, decision to publish, or preparation of the manuscript.

### Grant Disclosures

The following grant information was disclosed by the authors:
National Research Council of Thailand.
Office of Higher Education Commission: NRU-59-002-HR.
National Science and Technology Development Agency: P-15-50004.
Outstanding Professor of Thailand Research Fund: DPG5480002.
Chulalongkorn University Ratchadaphiseksomphot Endowment Fund: RES560530093.
Centenary Academic Development Project: CU56-HR01.
Center of Excellence in Clinical Virology: GCE 58-014-30-004.
King Chulalongkorn Memorial Hospital.
Rachadapisek Sompote Fund of Chulalongkorn University to Rujipat Wasitthankasem.
100th Anniversary Chulalongkorn University Fund.

## Competing Interests

The authors declare there are no competing interests.

## Author Contributions

- Ilada Thongpan conceived and designed the experiments, performed the experiments, analyzed the data, wrote the paper, prepared figures and/or tables, reviewed drafts of the paper.
- John Mauleekoonphairoj performed the experiments, analyzed the data.
- Preeyaporn Vichiwattana and Sumeth Korkong performed the experiments, contributed reagents/materials/analysis tools.
- Rujipat Wasitthankasem analyzed the data, reviewed drafts of the paper.
- Sompong Vongpunsawad analyzed the data, wrote the paper, reviewed drafts of the paper.
- Yong Poovorawan conceived and designed the experiments, wrote the paper, reviewed drafts of the paper.

## Human Ethics

The following information was supplied relating to ethical approvals (i.e., approving body and any reference numbers):

This study was approved by the Institutional Review Board of the Faculty of Medicine, Chulalongkorn University (IRB 609/59).

## DNA Deposition

The following information was supplied regarding the deposition of DNA sequences:

GenBank accession numbers KY327937–KY328054 (RSV-A) and KY328055–KY328148 (RSV-B).

## Data Availability

The raw data are located in the tables, figures, and Supplemental Information.

## Supplemental Information

Supplemental information for this article can be found online at http://dx.doi.org/10.7717/peerj.3970#supplemental-information.

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
