# Peer review of "Respiratory syncytial virus genotypes NA1, ON1, and BA9 are prevalent in Thailand, 2012–2015"

_PeerJ, doi:10.7717/peerj.3970_

## Round 0.1 · original submission · Major Revisions

· Academic Editor

Major Revisions

As part of the reviewing process, myself and the reviewers need to be able to see the sequences of the viruses in order to check your conclusions. At present these have not been released in GenBank. Please check they are publicly available prior to resubmission.

Reviewer 1 ·

Basic reporting

This manuscript provides additional information on RSV genotypes circulation in Southeast Asia Region. As the RSV molecular epidemiology data from developing countries especially from long term studies are still lacking, this manuscript contributes on the data needed to determine the global RSV circulation and transmission.
.
There are some drawbacks of this manuscript:
1. There are several grammatical errors and some sentences need to be reframing.
The flow of idea of the sentences within the paragraphs and between the
paragraphs are unclear. There are some repetitions of vocabulary used in the
sentences.

2. The structure of the manuscript should be revised based on the PeerJ standards.
2.1 The abstract should be revised as it did not follow the PeerJ standard.
2.2 There is no conclusion paragraph. As the PeerJ standard, the author should add
the conclusion section.
3. The title of this manuscript is too general. The authors may change it to the more
specific title which related to the contents of the article.
4. Figures and tables: Figure 1 needs a revision (left vertical axis ?)

Raw data is not provided. I am unable to verify the RSV sequences obtained from this study from NCBI website as they mentioned that these sequences have not yet been released.

Experimental design

1. The topic of this manuscript falls within the scope of the journal
2. Research questions should be more well-explained, as it seems inconsistent from the
abstract, introduction and the discussion.
3. Methodology:
3.1. The author should clarify the methodology regarding the samples as in methods
section (line 82) the samples were from a retrospective study while in the discussion
(line 1993) the author mentioned about multi-year surveillance study.
3.2. The author should add the screening diagnostic methods used for the samples as it is
mentioned in the abstract but it is not well-explained in the methods.
3.3. Detailed methodology on DNA sequencing should be provided.

Validity of the findings

1. The authors do not provide the demography data of the samples.
2. The discussion on the first and second results should include the comparison with the
results from the previous RSV studies in Thailand, Southeast Asia countries and
other countries in tropical region.
3. The authors should discuss the results of amino acid substitutions (line 161-181) as
the discussion is limited to N-linked glycosylation analysis.
4. The authors should provide more discussions on the third and fourth results. In the
discussion (line 239-258), it is not clear whether the authors discuss their results or
the results from the references.
5. As previously mentioned (point B.2.2), the authors should provide a comprehensive
conclusion section.

Additional comments

The authors should improve the manuscripts (language, grammar, and the clarity of the presentation) before publication. Furthermore, the authors should provide more detailed discussions on the results presented in the article. Overall, the authors should consider the manuscript to be reviewed and edited by a native English speaker to improve the clarity.

Reviewer 2 ·

Basic reporting

no comment

Experimental design

no comment

Validity of the findings

no comment

Additional comments

RSV infects lower respiratory track of, especially, children worldwide. In the current study, Thongpan et al. used molecular techniques to describe epidemiological features of RSV Thai strains, detected in several thousands samples, collected over the past couples of years (2013-5). The study is well described, and provides updated data on RSV infection in Thailand. I have some comments, below:

Line 29-30: It will be helpful to give a brief description of 'RSV-A' and 'RSV-B' in the abstract, for readers who are not familiar with these terms.

Line 33: Perhaps the results reported here could describe only in the context of Thailand, not Southeastern Asia.

Line 53: Should it be 13 RSV-A genotypes (not 11)?

Line 65: Any info to describe how the particular strain of virus did spread around the world?

Line 83-91: Were the 3,306 specimens derived from 3,306 subjects/patients? If not, from how many subjects/patients? How many samples were from subjects with flu-like illness and probable RSV infection? How many samples were from King Chulalongkorn Memorial Hospital and from Chum Phae Hospital? Such info should be mentioned in the manuscript.

Line 108: There were 278 positive samples. I wonder whether a total of 212 strains (118 RSV-A and 94 RSV-B) were identified from all these positive samples, or there were just some positive samples that failed to identify the strains?

Line 112: What are the genotypes of the RSV reference strains B1 and A2?

Line 171 and 176: duplication of which gene or region?

Line 211: It will be more interesting to see if there is any info about the percentages of co-infection by both RSV subgroups.

Figure 1: Is it possible to adjust the scale of the Y-axis to 0 - 100%, to clearer describe % positive samples, in relation to the number of samples, at any time points?

Figure 2 and 3: Can the Figures 2 and 3 be combined into one tree, to see overall phylogenetic relationships of all RSV Thai strains? In the legends, adding some info regarding about number of samples, where they were derived, how many samples in each year, and indicators of which are outgroup/reference strains, would be helpful to readers.

Figure 4 and 5: the symbols '.' and '~' and any others, marked in the figure, should be given a short description in the legend. Can these two figures be combined into one sequence alignment of both RSV subgroups?

---

## Round 0.2 · accepted · Accept

· Academic Editor

Accept

You have addressed all of the concerns from the reviewers adequately. At this stage could you now make sure that the accession numbers on GenBank are open and accessible.